# Individually Fair Ranking

**Amanda Bower**
Department of Mathematics
University of Michigan
amandarg@umich.edu

**Hamid Eftekhari**
Department of Statistics
University of Michigan
hamidef@umich.edu

**Mikhail Yurochkin**
IBM Research
MIT-IBM Watson AI Lab
mikhail.yurochkin@ibm.com

**Yuekai Sun**
Department of Statistics
University of Michigan
yuekai@umich.edu

## Abstract

We develop an algorithm to train individually fair learning-to-rank (LTR) models. The proposed approach ensures items from minority groups appear alongside similar items from majority groups. This notion of fair ranking is based on the definition of individual fairness from supervised learning and is more nuanced than prior fair LTR approaches that simply ensure the ranking model provides underrepresented items with a basic level of exposure. The crux of our method is an optimal transport-based regularizer that enforces individual fairness and an efficient algorithm for optimizing the regularizer. We show that our approach leads to certifiably individually fair LTR models and demonstrate the efficacy of our method on ranking tasks subject to demographic biases.

## 1 Introduction

Information retrieval (IR) systems are everywhere in today's digital world, and ranking models are integral parts of many IR systems. In light of their ubiquity, issues of algorithmic bias and unfairness in ranking models have come to the fore of the public's attention. In many applications, the items to be ranked are individuals, so algorithmic biases in the output of ranking models directly affect people's lives. For example, gender bias in job search engines directly affect the career success of job applicants (Dastin, 2018).

There is a rapidly growing literature on detecting and mitigating algorithmic bias in machine learning (ML). The ML community has developed many formal definitions of algorithmic fairness along with algorithms to enforce these definitions (Dwork et al., 2012; Hardt et al., 2016; Berk et al., 2018; Kusner et al., 2018; Ritov et al., 2017; Yurochkin et al., 2020). Unfortunately, these issues have received less attention in the IR community. In particular, compared to the myriad of mathematical definitions of algorithmic fairness in the ML community, there are only a few definitions of algorithmic fairness for ranking. A recent review of fair ranking (Castillo, 2019) identifies two characteristics of fair rankings:

1. sufficient exposure of items from disadvantaged groups in rankings: Rankings should display a diversity of items. In particular, rankings should take care to display items from disadvantaged groups to avoid allocative harms to items from such groups.
2. consistent treatment of similar items in rankings: Items with similar relevant attributes should be ranked similarly.

There is a line of work on fair ranking by Singh & Joachims (2018; 2019) that focuses on the first characteristic. In this paper, we complement this line of work by focusing on the second characteristic. In particular, we (i) specialize the notion of individual fairness in ML to rankings and (ii) devise an efficient algorithm for enforcing this notion in practice. We focus on the second characteristic since, in some sense, consistent treatment of similar items implies adequate exposure: if there are items from disadvantaged groups that are similar to relevant items from advantaged groups, then a ranking model that treats similar items consistently will provide adequate exposure to the items from disadvantaged groups.

## 1.1 RELATED WORK

Our work addresses the fairness of a learning-to-rank (LTR) system with respect to the items being ranked. The majority of work in this area requires a fair ranking to fairly allocate exposure (measured by the rank of an item in a ranking) to items. One line of work (Yang & Stoyanovich, 2017; Zehlike et al., 2017; Celis et al., 2018; Geyik et al., 2019; Celis et al., 2020; Yang et al., 2019b) requires a fair ranking to place a minimum number of minority group items in the top $k$ ranks. Another line of work models the exposure items receive based on rank position and allocates exposure based on these exposure models and item relevance (Singh & Joachims, 2018; Zehlike & Castillo, 2020; Biega et al., 2018; Singh & Joachims, 2019; Sapiezynski et al., 2019). There is some work that consider other fairness notions. The work of Kuhlman et al. (2019) proposes error-based fairness criteria, and the framework of Asudeh et al. (2019) can handle arbitrary fairness constraints given by an oracle. In contrast, we propose a fundamentally new definition: an individually fair ranking is invariant to sensitive perturbations of the features of the items. For example, consider ranking a set of job candidates, and consider the hypothetical set of candidates obtained from the original set by flipping each candidate's gender. We require that a fair LTR model produces the same ranking for both the original and hypothetical set.

The work in Zehlike et al. (2017); Celis et al. (2018); Singh & Joachims (2018); Biega et al. (2018); Geyik et al. (2019); Celis et al. (2020); Yang et al. (2019b); Wu et al. (2018); Asudeh et al. (2019) propose post-processing algorithms to obtain a fair ranking, i.e., algorithms that fairly re-rank items based on estimated relevance scores or rankings from potentially biased LTR models. However, post-processing techniques are insufficient since they can be mislead by biased estimated relevance scores (Zehlike & Castillo, 2020; Singh & Joachims, 2019) with the exception of the work in Celis et al. (2020) which assumes a specific bias model and provably counteracts this bias. In contrast, like Zehlike & Castillo (2020); Singh & Joachims (2019), we propose an in-processing algorithm. We also note that there is some work on

We consider individual fairness as opposed to group fairness (Yang & Stoyanovich, 2017; Zehlike et al., 2017; Celis et al., 2018; Singh & Joachims, 2018; Zehlike & Castillo, 2020; Geyik et al., 2019; Sapiezynski et al., 2019; Kuhlman et al., 2019; Celis et al., 2020; Yang et al., 2019b; Wu et al., 2018; Asudeh et al., 2019). The merits of individual fairness over group fairness have been well established, e.g., group fair models can be blatantly unfair to individuals (Dwork et al., 2012). In fact, we show empirically that individual fairness is adequate for group fairness but not vice versa. The work in Biega et al. (2018); Singh & Joachims (2019) also considers individually fair LTR models. However, our notion of individual fairness is fundamentally different since we utilize a fair metric on queries like in the seminal work that introduced individual fairness (Dwork et al., 2012) instead of measuring the similarity of items through relevance alone. To see the benefit of our approach, consider the job applicant example. If the training data does not contain highly ranked minority candidates, then at test time our LTR model will be able to correctly rank a minority candidate who should be highly ranked, which is not necessarily true for the work in Biega et al. (2018); Singh & Joachims (2019).

## 2 PROBLEM FORMULATION

A query $q \in \mathcal{Q}$ to a ranker consists of a candidate set of $n$ items that needs to be ranked $d^q \triangleq \{d_1^q, \ldots, d_n^q\}$ and a set of relevance scores $\text{rel}^q \triangleq \{\text{rel}^q(d) \in \mathbb{R}\}_{d \in d^q}$. Each item is represented by a feature vector $\varphi(d) \in \mathcal{X}$ that describes the match between item $d$ and query $q$ where $\mathcal{X}$ is the feature space of the item representations. We consider stochastic ranking policies $\pi(\cdot \mid q)$ that are distributions over rankings $r$ (*i.e.* permutations) of the candidate set. Our notation for rankings is $r(d)$: the rank of item $d$ in ranking $r$ (and $r^{-1}(j)$ is the $j$-ranked item). A policy generally consists of two components: a scoring model and a sampling method. The scoring model is a smooth ML model $h_\theta$ parameterized by $\theta$ (*e.g.* a neural network) that outputs a vector of scores: $h_\theta(\varphi(d^q)) \triangleq (h_\theta(\varphi(d_1^q)), \ldots, h_\theta(\varphi(d_n^q)))$. The sampling method defines a distribution on rankings of the candidate set from the scores. For example, the Plackett-Luce (Plackett, 1975) model defines the probability of the ranking $r = \langle d_1, \ldots, d_n \rangle$ as

$$\pi_\theta(r \mid q) = \prod_{j=1}^{n} \frac{\exp(h_\theta(\varphi(d_j)))}{\exp(h_\theta(\varphi(d_j))) + \cdots + \exp(h_\theta(\varphi(d_n)))}. \tag{2.1}$$

To sample a ranking from the Placket-Luce model, items from a query are chosen without replacement where the probability of selecting items is given by the softmax of the scores of remaining items. The order in which the items are sampled defines the order of the ranking from best to worst. The goal of the LTR problem is finding a policy that has maximum expected utility:

$$\pi^* \triangleq \arg\max_\pi \mathbb{E}_{q \sim Q}\big[U(\pi \mid q)\big] \text{ where } U(\pi \mid q) \triangleq \mathbb{E}_{r \sim \pi(\cdot \mid q)}\big[\Delta(r, \mathrm{rel}^q)\big], \tag{2.2}$$

where $Q$ is the distribution of queries, $U(\pi \mid q)$ is the utility of a policy $\pi$ for query $q$, and $\Delta$ is a ranking metric (*e.g.* normalized discounted cumulative gain). In practice, we solve the empirical version of (2.2):

$$\widehat{\pi} \triangleq \arg\max_\pi \frac{1}{N}\sum_{i=1}^{N}\big[U(\pi \mid q_i)\big], \tag{2.3}$$

where $\{q_i\}_{i=1}^N$ is a training set. If the policy is parameterized by $\theta$, it is not hard to evaluate the gradient of the utility with respect to $\theta$ with the log-derivative trick:

$$\partial_\theta U(\pi_\theta \mid q) = \partial_\theta \mathbb{E}_{r \sim \pi_\theta(\cdot \mid q)}\big[\Delta(r, \mathrm{rel}^q)\big] = \int \Delta(r, \mathrm{rel}^q)\partial_\theta \pi_\theta(r \mid q)dr$$

$$= \int \Delta(r, \mathrm{rel}^q)\partial_\theta\{\log \pi_\theta(r \mid q)\}\pi_\theta(r \mid q)dr = \mathbb{E}_{r \sim \pi_\theta(\cdot \mid q)}\big[\Delta(r, \mathrm{rel}^q)\partial_\theta \log \pi_\theta(r \mid q)\big].$$

In practice, we (approximately) evaluate $\partial_\theta U(\pi_\theta \mid q)$ by sampling from $\pi_\theta(\cdot \mid q)$. This set-up is mostly adopted from Yadav et al. (2019).

## 2.1 FAIR RANKING VIA INVARIANCE REGULARIZATION

We cast the fair ranking problem as training ranking policies that are invariant under certain sensitive perturbations to the queries. Let $d_Q$ be a fair metric on queries that encode which queries should be treated similarly by the LTR model. For example, a LTR model should similarly rank a set of job candidates and the hypothetical set of job candidates obtained from the original set via flipping the gender of each candidate. Hence, these two queries should be close according to $d_Q$. We propose Sensitive Set Transport Invariant Ranking (SenSTIR) to enforce individual fairness in ranking via the following optimization problem:

$$\pi^* \triangleq \arg\max_\pi \mathbb{E}_{q \sim Q}\big[U(\pi \mid q)\big] - \rho R(\pi), \tag{SenSTIR}$$

such that $\rho > 0$ is a regularization parameter and

$$R(\pi) \triangleq \left\{ \begin{array}{ll} \sup_{\Pi \in \Delta(Q \times Q)} & \mathbb{E}_{(q,q') \sim \Pi}\big[d_{\mathcal{R}}(\pi(\cdot \mid q), \pi(\cdot \mid q'))\big] \\ \text{subject to} & \mathbb{E}_{(q,q') \sim \Pi}\big[d_Q(q, q')\big] \leq \epsilon \\ & \Pi(\cdot, Q) = Q \end{array} \right\} \tag{2.4}$$

is an invariance regularizer where $d_{\mathcal{R}}$ is a metric on ranking policies, $\Delta(Q \times Q)$ is the set of probability distributions on $Q \times Q$ where $Q$ is the set of queries, and $\epsilon > 0$. At a high-level, individual fairness requires ML models to have similar outputs for similar inputs. This property is exactly what the regularizer encourages: the LTR model is encouraged to assign similar ranking policies (with respect to $d_{\mathcal{R}}$) to similar queries (with respect to $d_Q$). The problem of enforcing invariance for individual fairness has been considered in classification (Yurochkin et al., 2020; Yurochkin & Sun, 2021). However, these methods are not readily applicable to the LTR setting because of two main challenges: (i) defining a fair distance $d_Q$ on queries, i.e., *sets* of items, and (ii) ensuring the resulting optimization problem is differentiable.

**Optimal transport distance $d_Q$ between queries**  We appeal to the machinery of optimal transport to define an appropriate metric $d_Q$ on queries, i.e., *sets* of items. First, we need a fair metric on items $d_{\mathcal{X}}$ that encodes our intuition of which items should be treated similarly. Such a metric also appears in the traditional individual fairness definition (Dwork et al., 2012) for classification and regression problems. Learning an individually fair metric is an important problem of its own that is actively studied in the recent literature (Ilvento, 2020; Wang et al., 2019; Yurochkin et al., 2020; Mukherjee et al., 2020). In the experiment section, the fair metric on items $d_{\mathcal{X}}$ is learned from data

using existing methods. The key idea is to view queries, i.e., *sets* of items, as distributions on $\mathcal{X}$ so that a metric between distributions can be used. In particular, to define $d_{\mathcal{Q}}$ from $d_{\mathcal{X}}$, we utilize an optimal transport distance between queries with $d_{\mathcal{X}}$ as the transport cost:

$$d_{\mathcal{Q}}(q,q') \triangleq \begin{cases} \inf_{\Pi \in \Delta(\mathcal{X} \times \mathcal{X})} & \int_{\mathcal{X} \times \mathcal{X}} d_{\mathcal{X}}(x,x') d\Pi(x,x') \\ \text{subject to} & \Pi(\cdot, \mathcal{X}) = \frac{1}{n} \sum_{j=1}^n \delta_{\varphi(d_j^q)} \\ & \Pi(\mathcal{X}, \cdot) = \frac{1}{n} \sum_{j=1}^n \delta_{\varphi(d_j^{q'})} \end{cases} , \qquad (2.5)$$

where $\Delta(\mathcal{X} \times \mathcal{X})$ is the set of probability distributions on $\mathcal{X} \times \mathcal{X}$ where $\mathcal{X}$ is the feature space of item representations and $\delta$ is the Dirac delta function.

## 3 ALGORITHM

In order to apply stochastic optimization to Equation (SenSTIR), we appeal to duality. In particular, we use Theorem 2.3 of Yurochkin & Sun (2021) re-written with the notation of this work:

**Theorem** (Theorem 2.3 of Yurochkin & Sun (2021)). *If $d_{\mathcal{R}}(\pi(\cdot \mid q), \pi(\cdot \mid q')) - \lambda d_{\mathcal{Q}}(q,q')$ is continuous in $(q,q')$ for all $\lambda$, then the invariance regularizer $R$ can be written as*

$$R(\pi) = \inf_{\lambda \geq 0} \{ \lambda \epsilon + \mathbb{E}_{q \sim Q}[r_\lambda(\pi, q)] \}, \text{where} \qquad (3.1)$$

$$r_\lambda(\pi, q) \triangleq \sup_{q' \in \mathcal{Q}} \{ d_{\mathcal{R}}(\pi(\cdot \mid q), \pi(\cdot \mid q')) - \lambda d_{\mathcal{Q}}(q,q') \}. \qquad (3.2)$$

In order to compute $r_\lambda(\pi, q)$, we can use gradient ascent on $u(q' \mid \pi, q, \lambda) \triangleq d_{\mathcal{R}}(\pi(\cdot \mid q), \pi(\cdot \mid q')) - \lambda d_{\mathcal{Q}}(q,q')$. We start by computing the gradient of $d_{\mathcal{Q}}(q,q')$ with respect to $x' \triangleq \varphi(d^{q'})$. Let $x \triangleq \varphi(d^q)$. Let $\Pi^\star(q,q')$ be the optimal transport plan for the problem defining $d_{\mathcal{Q}}(q,q')$, that is

$$d_{\mathcal{Q}}(q,q') = \int_{\mathcal{X} \times \mathcal{X}} d_{\mathcal{X}}(x,x') d\Pi^\star(x,x'), \Pi^\star(\cdot, \mathcal{X}) = \frac{1}{n} \sum_{j=1}^n \delta_{\varphi(d_j^q)}, \Pi^\star(\mathcal{X}, \cdot) = \frac{1}{n} \sum_{j=1}^n \delta_{\varphi(d_j^{q'})}.$$

The probability distribution $\Pi^\star(q,q')$ can be viewed as a coupling matrix where $\Pi_{i,j}^\star \triangleq \Pi^\star(\varphi(d_i^q), \varphi(d_j^{q'}))$. Using this notation we have

$$\partial_{x_j'} d_{\mathcal{Q}}(q,q') = \sum_{i=1}^n \Pi_{i,j}^\star \partial_2 d_{\mathcal{X}}(\varphi(d_i^q), \varphi(d_j^{q'})), \qquad (3.3)$$

where $\partial_2 d_{\mathcal{X}}$ denotes the derivative of $d_{\mathcal{X}}$ with respect to its second input. If $d_{\mathcal{R}}(\pi_\theta(\cdot \mid q), \pi_\theta(\cdot \mid q')) = \|h_\theta(\varphi(d^q)) - h_\theta(\varphi(d^{q'}))\|_2^2/2$, then by (3.3), a single iteration of gradient ascent on $d_{\mathcal{Q}}$ with step size $\gamma$ for $x'$ is

$$x_j'^{(l+1)} = x_j'^{(l)} + \gamma \left( \partial_{x_j'} h_\theta(x'^{(l)})^T (h_\theta(x'^{(l)}) - h_\theta(x)) - \lambda \sum_{i=1}^n \Pi_{i,j}^\star \partial_2 d_{\mathcal{X}}(x_i, x_j'^{(l)}) \right). \qquad (3.4)$$

In our experiments, we use this choice of $d_{\mathcal{R}}$, which has been widely used, e.g., robustness in image classification (Kannan et al., 2018; Yang et al., 2019a) and fairness (Yurochkin & Sun, 2021). However, our theory and set-up do not preclude other metrics. We can now present Algorithm 1, an alternating, stochastic algorithm, to solve (SenSTIR).

---

**Algorithm 1:** SenSTIR: Sensitive Set Transport Invariant Ranking

**Input:** Initial Parameters: $\theta_0, \lambda_0, \epsilon, \rho$; Step Sizes: $\gamma, \alpha_t, \eta_t > 0$, Training queries: $\hat{Q}$

1 **repeat**

2     Sample mini-batch $(q_{t_i}, \text{rel}^{q_{t_i}})_{i=1}^B$ from $\hat{Q}$

3     $q_{t_i}' \leftarrow \arg\max_{q'} \{ \frac{1}{2} \| h_{\theta_t}(\varphi(d^{q_{t_i}})) - h_{\theta_t}(\varphi(d^{q'})) \|_2^2 - \lambda_t d_{\mathcal{Q}}(q_{t_i}, q') \}, i \in [B]$    /* Using (3.4) */

4     $\lambda_{t+1} \leftarrow \max\{0, \lambda_t + \alpha_t \rho(\epsilon - \frac{1}{B} \sum_{i=1}^B d_{\mathcal{Q}}(q_{t_i}, q_{t_i}'))\}$

5     $\theta_{t+1} \leftarrow \theta_t + \eta_t (\frac{1}{B} \sum_{i=1}^B \partial_\theta \{ U(\pi_{\theta_t} \mid q_{t_i}) \} - \rho(\partial_\theta h_{\theta_t}(q_{t_i}') - \partial_\theta h_{\theta_t}(q_{t_i}))^T (h_{\theta_t}(q_{t_i}') - h_{\theta_t}(q_{t_i}))$

6 **until** convergence

---

## 4 THEORETICAL RESULTS

In this section, we study the generalization performance of the invariance regularizer $R(h_\theta) := R(\pi_\theta)$, which is an instance of a hierarchical optimal transport problem that does not have known uniform convergence results in the literature. Furthermore, the regularizer is not a separable function of the training examples so classical proof techniques are not applicable. To state the result, suppose that $\hat{d}_\mathcal{X}$ is an approximation of the fair metric $d_\mathcal{X}$ between items that is learned from data. The corresponding learned metric on queries is defined by

$$
\hat{d}_\mathcal{Q}(q,q') \triangleq \begin{cases} \inf_{\Pi \in \Delta(\mathcal{X} \times \mathcal{X})} & \int_{\mathcal{X} \times \mathcal{X}} \hat{d}_\mathcal{X}(x,x') d\Pi(x,x') \\ \text{subject to} & \Pi(\cdot, \mathcal{X}) = \frac{1}{n} \sum_{j=1}^n \delta_{\varphi(d_j^q)} \\ & \Pi(\mathcal{X}, \cdot) = \frac{1}{n} \sum_{j=1}^n \delta_{\varphi(d_j^{q'})} \end{cases}, \tag{4.1}
$$

and the empirical regularizer is defined by

$$
\hat{R}(h_\theta) \triangleq \begin{cases} \sup_{\Pi \in \Delta(\mathcal{Q} \times \mathcal{Q})} & \mathbb{E}_\Pi \left[ d_\mathcal{Y}(h_\theta(\varphi(d^q)), h_\theta(\varphi(d^{q'}))) \right] \\ \text{subject to} & \mathbb{E}_\Pi \left[ \hat{d}_\mathcal{Q}(q,q') \right] \le \epsilon \\ & \Pi(\cdot, \mathcal{Q}) = \hat{Q} \end{cases}, \tag{4.2}
$$

where $\hat{Q}$ is the distribution of training queries and $d_\mathcal{Y}$ is a metric on $\mathcal{Y} \triangleq \{h_\theta(\varphi(d^q)) \mid q \in \mathcal{Q}\}$.

Define a class of loss functions $\mathcal{D}$ by $\mathcal{D} \triangleq \{d_{h_\theta} : \mathcal{Q} \times \mathcal{Q} \to \mathbf{R}_+ \mid h_\theta \in \mathcal{H}\}$, where $d_h(q,q') \triangleq d_\mathcal{Y}(h(\varphi(d^q)), h(\varphi(d^{q'})))$ and $\mathcal{H}$ is the hypothesis class of scoring functions.

Let $N(\mathcal{D}, d, \epsilon)$ be the $\epsilon$-covering of the class $\mathcal{D}$ with respect to a metric $d$. The entropy integral of $\mathcal{D}$ (w.r.t. the uniform metric) measures the complexity of the class and is defined by

$$
J(\mathcal{D}) \triangleq \int_0^\infty \sqrt{\log N(\mathcal{D}, \|\cdot\|_\infty, \epsilon)} d\epsilon. \tag{4.3}
$$

**Assumption A1.** Bounded diameters: $\sup_{x,x' \in \mathcal{X}} d_\mathcal{X}(x,x') \le D_\mathcal{X}$, $\sup_{y,y' \in \mathcal{Y}} d_\mathcal{Y}(y,y') \le D_\mathcal{Y}$.

**Assumption A2.** Estimation error of $d_\mathcal{X}$ is bounded: $\sup_{x,x' \in \mathcal{X}} |\hat{d}_\mathcal{X}(x,x') - d_\mathcal{X}(x,x')| \le \eta_d$.

**Theorem 4.1.** *If assumptions A1 and A2 hold and $J(\mathcal{D})$ is finite, then with probability at least $1 - t$*

$$
\sup_{h_\theta \in \mathcal{H}} |\hat{R}(h_\theta) - R(h_\theta)| \le \frac{48(J(\mathcal{D}) + \epsilon^{-1} D_\mathcal{X} D_\mathcal{Y})}{\sqrt{n}} + D_\mathcal{Y} \left( \frac{\log \frac{2}{t}}{2n} \right)^{\frac{1}{2}} + \frac{D_\mathcal{Y} \eta_d}{\epsilon},
$$

where $n$ is the number of training queries. A proof of the theorem is given in the appendix. The key technical challenge is leveraging the transport geometry on the query space to obtain a uniform bound on the convergence rate. This theorem implies that for a trained ranking model $\hat{h}_\theta$, the error term $|\hat{R}(\hat{h}_\theta) - R(\hat{h}_\theta)|$ is small for large $n$. Therefore, one can certify that the value of the regularizer $R(\hat{h}_\theta)$ is small on yet unseen (test) data by ensuring that the value of $\hat{R}(\hat{h}_\theta)$ is small on training data.

## 5 COMPUTATIONAL RESULTS

In this section, we demonstrate the efficacy of SenSTIR for learning individually fair LTR models. One key conclusion is that enforcing individual fairness is adequate to achieve group fairness but not vice versa. See Section B of the appendix for full details about the experiments.

**Fair metric** Following Yurochkin et al. (2020), the individually fair metric $d_\mathcal{X}$ on $\mathcal{X}$ is defined in terms of a *sensitive subspace* $A$ that is learned from data. In particular, $d_\mathcal{X}$ is the Euclidean distance of the data projected onto the orthogonal complement of $A$. This metric encodes variation due to sensitive information about individuals in the subspace and ignores it when computing the fair distance. For example, $A$ can be formed by fitting linear classifiers to predict sensitive information,

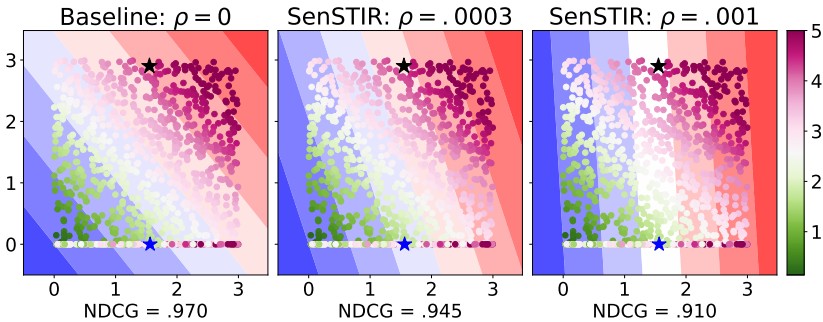

Figure 1: The points represent items shaded by their relevances, and the contours represent the predicted scores. The minority items lie on the horizontal $z_1$-axis because their $z_2$ value is corrupted to 0. The blue star and black star correspond to minority and majority items that are close in the fair metric with nearly the same relevance. However, they have wildly different predicted scores under the baseline. Using SenSTIR, as $\rho$ increases, they eventually have the same predicted scores.

like gender or age, of individuals and taking the span of the vectors orthogonal to the corresponding decision boundaries. In each experiment, we explain how $A$ is learned.

**Baselines** For all methods, we learn linear score functions $h_\theta$ and maximize normalized discounted cumulative gain (NDCG), i.e., $\Delta$ in Equation 2.2 is NDCG. We compare SenSTIR to (1) vanilla training without fairness ("Baseline"), i.e., $\rho = 0$, (2) pre-processing by first projecting the data onto the orthogonal complement of the sensitive subspace and then using vanilla training ("Project"), (3) "Fair-PG-Rank" (Singh & Joachims, 2019), a recent approach for training fair LTR models, and (4) randomly sampling the linear weights from a standard normal ("Random") to give context to NDCG.

## 5.1 SYNTHETIC

We use synthetic data considered in prior fair ranking work (Singh & Joachims, 2019). Each query contains 10 majority or minority items in $\mathbb{R}^2$ such that 8 items per query are majority group items in expectation. For each item, $z_1$ and $z_2$ are drawn uniformly from $[0, 3]$. The relevance of an item is $z_1 + z_2$ clipped between 0 and 5. A majority item's feature vector is $(z_1, z_2)^T$, whereas a minority item's feature vector is corrupted and given by $(z_1, 0)^T$.

**Fair Metric** The sensitive subspace is spanned by the hyperplane learned by logistic regression to predict whether an item is in the majority group. Recall, the fair metric is the Euclidean distance of the projection of the data onto the orthogonal complement of this subspace. Since this hyperplane is nearly equal to $(0, 1)^T$, the biased feature $z_2$ is ignored in the fair metric.

**Results** Figure 1 illustrates SenSTIR for $\rho \in \{0, .0003, .001\}$ with $\epsilon = .001$. Each point is colored by its relevance, and the contours show predicted scores where redder (respectively bluer) regions indicates higher (respectively lower) predicted scores. Minority items are on the horizontal $z_1$-axis because of their corrupted features. When $\rho = 0$, i.e., fairness is not enforced, this score function badly violates individual fairness since there are pairs of items close in the fair metric but with wildly different predicted scores because the biased feature $z_2$ is used. For example, the bottom blue star is a minority item with nearly the same relevance as the top black star majority item; however, the majority item's predicted score is much higher. When $\rho$ is increased, the contours learned by SenSTIR eventually become vertical, thereby ignoring the biased feature $z_2$ and achieving individual fairness. When $\rho = .001$, the scores of the blue and black star are nearly equal because they are very close in the fair metric and the fair regularization strength is large enough.

Figure 2 illustrates another individual fairness property of SenSTIR that Fair-PG-Rank does not satisfy: ranking stability with respect to sensitive perturbations of the features. For each test query $q$, let $q' \neq q$ be the closest test query in terms of the fair distance $d_\mathcal{Q}$. We can view $q'$ as a hypothetical query in the test set. For each query $q$, we sample 10 rankings corresponding to $q$ and 10 hypothetical rankings corresponding to $q'$ based on the learned ranking policy. The $(i, j)$-th entry of a heatmap in Figure 2 is the proportion of times the $i$-th ranked item for query $q$ is ranked $j$-th in the hypothetical

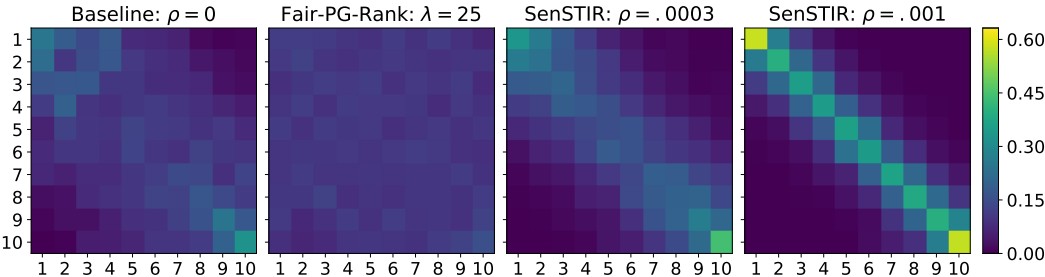

Figure 2: The $(i, j)$-th entries of these heatmaps represent the proportion of times that the $i$-th ranked item is moved to position $j$ under the corresponding hypothetical ranking. With large enough $\rho$, SenSTIR ranks the original queries and hypothetical queries similarly as desired.

ranking. To satisfy individual fairness, the original and hypothetical rankings should be similar, meaning the heatmaps should be close to diagonal. Even though the baseline is relatively stable for highly and lowly ranked items, these items still change positions under the hypothetical rankings more than 50% of the time. Although Fair-PG-Rank satisfies group fairness, it is worse than the baseline in terms of hypothetical stability, i.e., individual fairness. In contrast, as $\rho$ increases, SenSTIR becomes stable.

## 5.2 GERMAN CREDIT DATA SET

Following Singh & Joachims (2019), we adapt the German Credit classification data set (Dua & Graff, 2017), which is susceptible to gender and age biases, to a LTR task. This data set contains 1000 individuals with binary labels indicating creditworthiness. Features include demographics like gender and age as well as information about savings accounts, housing, and employment. To simulate LTR data, individuals are sampled with replacement to build queries of size 10. Each individual has a binary relevance, and on average 4 individuals are relevant in each query. To apply Fair-PG-Rank, age is the binary protected attribute where the two groups are those younger than 25 and those 25 and older, a split proposed by Kamiran & Calders (2009). For the fair metric, the sensitive subspace is spanned by the ridge regression coefficients for predicting age based on all other features and the standard basis vector corresponding to age.

**Comparison metrics** See Section B of the appendix for the precise definitions of these metrics. To assess accuracy, following Singh & Joachims (2019), we report the average stochastic test NDCG by sampling 25 rankings for each query from the learned ranking policy. To assess individual fairness, we use ranking stability with respect to demographic perturbations, which is the natural analogue of an evaluation metric for individual fairness in classification (Yurochkin & Sun, 2021; Yurochkin et al., 2020; Garg et al., 2018). In particular, for each query, we create a hypothetical query by flipping the (binary) gender of each individual in the query, and deterministically rank by sorting the items by their scores. We report the average Kendall's tau correlation (higher implies better individual fairness) between a test query's ranking and its hypothetical ranking. To assess group fairness and fairly compare to Fair-PG-Rank based on their fairness definition, we report the average stochastic disparity of group exposure also with 25 sampled rankings per query. This metric measures the asymmetric differences of the ratio of exposure a group receives to its relevance per query and favors the group with less relevance for a given query. Let $G_1$ (respectively $G_0$) be the set of older (respectively younger) people for a query $q$. For $i \in \{0, 1\}$, let $M_{G_i} = (1/|G_i|) \sum_{d \in G_i} \text{rel}^q(d)$. If $M_{G_0} > M_{G_1}$, let $G_A = G_0$, $G_D = G_1$ and $G_A = G_1$, $G_D = G_0$ otherwise. The stochastic disparity of group exposure for a set of rankings $\{r_i\}_{i=1}^{N}$ corresponding to a query is

$$\max \left\{ 0, \frac{\frac{1}{N|G_A|} \sum_{d \in G_A} \sum_{i=1}^{N} \frac{1}{\log_2(r_i(d)+1)}}{M_{G_A}} - \frac{\frac{1}{N|G_D|} \sum_{d \in G_D} \sum_{i=1}^{N} \frac{1}{\log_2(r_i(d)+1)}}{M_{G_D}} \right\}. \quad (5.1)$$

**Results** Figure 3 illustrates the fairness versus accuracy trade-off on the test set. The error bars represent the standard error over 10 random train/test splits. Both SenSTIR and Fair-PG-Rank enforce fairness through regularization, so we vary the regularization strength ($\rho$ for SenSTIR with $\epsilon$ constant). Based on the NDCG of "Random", the regularization strength ranges are reasonable for both methods. The left plot in Figure 3 shows the average Kendall's tau correlation (higher is better)

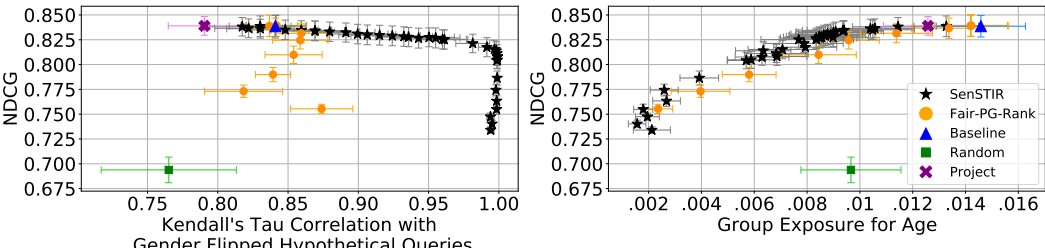

Figure 3: Individual (left) and group fairness (right) versus accuracy for the German credit data set

between test queries and their gender-flipped hypotheticals versus the average stochastic NDCG. The maximum Kendall's tau correlation is 1, which SenSTIR achieves with relatively high NDCG. We emphasize that the sensitive subspace that SenSTIR utilizes to define the fair query metric directly relates to age, not gender. In other words, our goal is to mitigate unfairness that arises from *age* in the training data, not *gender*. However, age is correlated with gender, so this metric shows the individually fair properties of SenSTIR generalize beyond age on the test set. We imagine that ML systems can be unfair to people with respect to features that can be difficult to know before deploying these systems, so although flipping gender is a simplistic choice, it illustrates that SenSTIR can be meaningfully individually fair with respect to these potentially unknown features that were not given special consideration when choosing the fair metric or in training with SenSTIR. Furthermore, SenSTIR gracefully trades off NDCG for individual fairness unlike Fair-PG-Rank. "Project" is worse in terms of individual fairness than vanilla training without enforcing fairness. Without direct age information, perhaps "Project" must more heavily rely on gender to learn accurate rankings, which illustrates that SenSTIR's generalization properties from age to gender are non-trivial. Disparity of group exposure (where smaller numbers are better) versus NDCG is depicted on the right plot of Figure 3. This group fairness metric is exactly what Fair-PG-Rank regularizes with. On average, for the same value of NDCG, SenSTIR typically outperforms Fair-PG-Rank showing that individual fairness can be adequate for group fairness but not vice versa. While "Project" improves mildly upon the baseline, it shows being "age" blind does not result in group fair rankings.

## 5.3 MICROSOFT LEARNING TO RANK DATA SET

The demographic biases are real in the German Credit data, but the LTR task is simulated. There are no standard LTR data sets with demographic biases, so we consider Microsoft's Learning to Rank (MSLR) data set (Qin & Liu, 2013) with an artificial algorithmic fairness concern dealing with webpage quality following Yadav et al. (2019). The data set consists of query-web page pairs from a search engine with nearly 140 features with integral relevance scores. To apply Fair-PG-Rank, following Yadav et al. (2019), the protected binary attribute is whether a web page is high or low quality defined by the 40th percentile of quality scores (feature 133). For the fair metric, the sensitive subspace is spanned by the ridge regression coefficients for predicting the quality score (feature 133) based on all features and the standard basis vector corresponding to the quality score.

**Comparison metrics** Again we use average stochastic NDCG to measure accuracy, and the dispartiy of group exposure where the groups are high and low quality web pages. To assess individual fairness, we use the same set-up as in the German Credit experiments except the hypothetical for each test query $q$ is the closest query $q' \neq q$ with respect to the fair metric over the train and test set.

**Results** Figure 4 shows the fairness and accuracy trade-off on the test set. Fair-PG-Rank becomes unstable with large fair regularization as it can drop below a random ranking in NDCG. The left plot shows the Kendall's tau correlation between test queries and their hypotheticals. SenSTIR gracefully trades-off NDCG with Kendall's tau correlation unlike Fair-PG-Rank. The right plot shows that SenSTIR also smoothly trades-off group fairness for NDCG. In contrast, as the regularization strength increases, both NDCG and group exposure worsen for Fair-PG-Rank, which was also observed by Yadav et al. (2019).

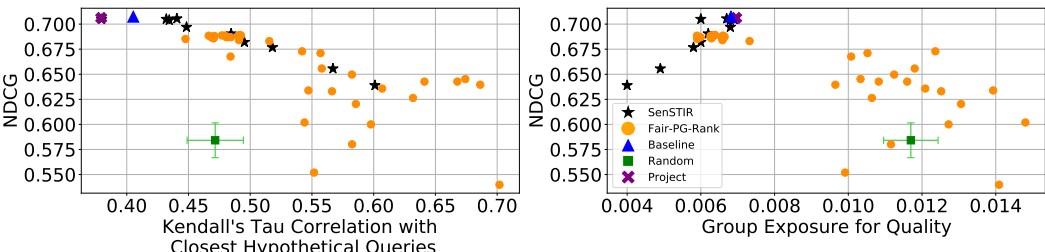

Figure 4: Individual (left) and group fairness (right) versus NDCG for the MSLR data set

## 6 CONCLUSION

We proposed SenSTIR, an algorithm to learn provably individually fair LTR models with an optimal transport-based regularizer. This regularizer encourages the LTR model to produce similar ranking policies, i.e., distributions over rankings, for similar queries where similarity is defined by a fair metric. Our notion of a fair ranking is complementary to prior definitions that require allocating exposure to items fairly with respect to merit. In fact, we empirically showed that enforcing individual fairness can lead to allocating exposure fairly for groups but allocating exposure fairly for groups does not necessarily lead to individually fair LTR models. An interesting future work direction is studying the fairness of LTR systems in the context of long-term effects (Mladenov et al., 2020).

## ACKNOWLEDGEMENTS

This paper is based upon work supported by the National Science Foundation (NSF) under grants no. 1830247 and 1916271. Any opinions, findings, and conclusions or recommendations expressed in this paper are those of the authors and do not necessarily reflect the views of the NSF.

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

## A    PROOFS OF THEORETICAL RESULTS

**Theorem A.1** (Theorem 4.1). *If assumptions A1 and A2 hold and $J(\mathcal{D})$ is finite, then with probability at least $1 - t$*

$$\sup_{h \in \mathcal{H}} |\hat{R}(h) - R(h)| \leq \frac{48(J(\mathcal{D}) + \varepsilon^{-1} D_{\mathcal{X}} D_{\mathcal{Y}})}{\sqrt{n}} + D_{\mathcal{Y}} \left( \frac{\log \frac{2}{t}}{2n} \right)^{\frac{1}{2}} + \frac{D_{\mathcal{Y}} \eta_d}{\varepsilon}.$$

*Proof.* For queries $q, q'$ let

$$\Delta(q, q') = \{\Pi \in \Delta(\mathcal{X} \times \mathcal{X}) : \Pi(\mathcal{X}, \cdot) = \frac{1}{n} \sum_{j=1}^{n} \delta_{\varphi(d_j^q)}, \Pi(\cdot, \mathcal{X}) = \frac{1}{n} \sum_{j=1}^{n} \delta_{\varphi(d_j^{q'})}\}.$$

Let $\Pi^* \in \arg\min_{\Pi \in \Delta(q,q')} \mathbb{E}_{\Pi}[d_{\mathcal{X}}(X, X')]$ and observe that by assumption A2 and the definition of $d_Q$ and $\hat{d}_Q$ we have

$$\begin{aligned}
\hat{d}_{\mathcal{Q}}(q, q') - d_{\mathcal{Q}}(q, q') &= \inf_{\Pi \in \Delta(q,q')} \mathbb{E}_{\Pi}[\hat{d}_{\mathcal{X}}(X, X')] - \inf_{\Pi \in \Delta(q,q')} \mathbb{E}_{\Pi}[d_{\mathcal{X}}(X, X')] \\
&= \inf_{\Pi \in \Delta(q,q')} \mathbb{E}_{\Pi}[\hat{d}_{\mathcal{X}}(X, X')] - \mathbb{E}_{\Pi^*}[d_{\mathcal{X}}(X, X')] \\
&\leq \mathbb{E}_{\Pi^*}[\hat{d}_{\mathcal{X}}(X, X')] - \mathbb{E}_{\Pi^*}[d_{\mathcal{X}}(X, X')] \\
&= \mathbb{E}_{\Pi^*}[\hat{d}_{\mathcal{X}}(X, X') - d_{\mathcal{X}}(X, X')] \\
&\leq \eta_d.
\end{aligned}$$

Similarly,

$$d_Q(q, q') - \hat{d}_Q(q, q') \leq \mathbb{E}_{\hat{\Pi}^*}[d_{\mathcal{X}}(X, X') - \hat{d}_{\mathcal{X}}(X, X')] \leq \eta_d.$$

It follows that

$$|\hat{d}_Q(q, q') - d_Q(q, q')| \leq \eta_d. \tag{A.1}$$

Next, we will bound the difference $|\hat{R}(h) - R(h)|$. To lighten the notation, we write $h, h'$ for $h = h(\phi(d^q)), h' = h(\phi(d^{q'}))$. From the dual representation of $R(h)$ and $\hat{R}(h)$ we have

$$\hat{R}(h) - R(h) = \inf_{\lambda \geq 0} \{\lambda \epsilon + \mathbb{E}_{q \sim \hat{Q}}[\hat{r}_{\lambda}(h, q)]\} - \inf_{\lambda \geq 0} \{\lambda \epsilon + \mathbb{E}_{q \sim Q}[r_{\lambda}(h, q)]\} \tag{A.2}$$

$$= \inf_{\lambda \geq 0} \{\lambda \epsilon + \mathbb{E}_{q \sim \hat{Q}}[\hat{r}_{\lambda}(h, q)]\} - \lambda^* \epsilon - \mathbb{E}_{q \sim \hat{Q}}[\hat{r}_{\lambda^*}(h, q)] \tag{A.3}$$

$$\leq \mathbb{E}_{q \sim \hat{Q}}[\hat{r}_{\lambda^*}(h, q)] - \mathbb{E}_{q \sim Q}[r_{\lambda^*}(h, q)] \tag{A.4}$$

$$= \mathbb{E}_{q \sim \hat{Q}}[r_{\lambda^*}(h, q)] - \mathbb{E}_{q \sim Q}[r_{\lambda^*}(h, q)] + \mathbb{E}_{q \sim \hat{Q}}[\hat{r}_{\lambda^*}(h, q) - r_{\lambda^*}(h, q)]. \tag{A.5}$$

To bound the last term, note that

$$|\hat{r}_{\lambda^*}(h, q) - r_{\lambda^*}(h, q)| = \sup_{q'}\{\hat{d}_{\mathcal{Y}}(h, h') - \lambda^* d_{\mathcal{Q}}(q, q')\} - \sup_{q'}\{\hat{d}_{\mathcal{Y}}(h, h') - \lambda^* d_{\mathcal{Q}}(q, q')\} \tag{A.6}$$

$$\leq \lambda^* \sup_{q'}\{|d_{\mathcal{Q}}(q, q') - \hat{d}_{\mathcal{Q}}(q, q')| \tag{A.7}$$

$$\leq \lambda^* \eta_d. \tag{A.8}$$

Combining (A.8) and (A.5) yields

$$\hat{R}(h) - R(h) \leq \mathbb{E}_{q \sim \hat{Q}}[r_{\lambda^*}(h, q)] - \mathbb{E}_{q \sim Q}[r_{\lambda^*}(h, q)] + \lambda^* \eta_d. \tag{A.9}$$

Using a similar argument,

$$R(h) - \hat{R}(h) \leq \mathbb{E}_{q \sim Q}[r_{\hat{\lambda}^*}(h, q)] - \mathbb{E}_{q \sim \hat{Q}}[r_{\hat{\lambda}^*}(h, q)] + \hat{\lambda}^* \eta_d. \tag{A.10}$$

To find an upper bound on $\lambda^*$, observe that $r_\lambda(h, q) \geq 0$ for all $h \in \mathcal{H}, \lambda \geq 0$, as

$$r_\lambda(h, q) = \sup_{q' \in \mathcal{X}} \{d_\mathcal{Y}(h, h') - \lambda d_\mathcal{Q}(q, q')\}$$
$$\geq d_\mathcal{Y}(h, h) - \lambda d_\mathcal{Q}(q, q) = 0.$$

Thus

$$\lambda^* \varepsilon \leq \lambda^\star \varepsilon + \mathbb{E}_{q \sim \mathcal{Q}}[r_\lambda(h, q)] = R(h) \leq D_\mathcal{Y}.$$

Rearranging the above yields $\lambda^* \leq \frac{D_\mathcal{Y}}{\varepsilon}$ and the same upper bound is also valid for $\hat{\lambda}^\star$ by the same argument.

Combining inequalities (A.9,A.10) and the bound on $\lambda^*, \hat{\lambda}^*$, we can write

$$|\hat{R}(h) - R(h)| \leq \sup_{f \in \mathcal{F}} \left| \mathbb{E}_{q \sim \hat{Q}} f(q) - \mathbb{E}_{q \sim Q} f(q) \right| + \frac{D_\mathcal{Y} \eta_d}{\varepsilon},$$

where $\mathcal{F} = \{r_\lambda(h, \cdot) : \lambda \in [0, L], \ h \in \mathcal{H}\}$. A standard concentration argument proves

$$\sup_{f \in \mathcal{F}} \left| \mathbb{E}_{q \sim \hat{Q}} f(q) - \mathbb{E}_{q \sim Q} f(q) \right| \leq \frac{48(J(\mathcal{D}) + \varepsilon^{-1} D_\mathcal{X} D_\mathcal{Y})}{\sqrt{n}} + D_\mathcal{Y} (\frac{\log \frac{2}{t}}{2n})^{\frac{1}{2}}$$

with probability at least $1 - t$. This completes the proof of the theorem. $\square$

The main technical novelty in this proof is the bound on $\lambda_*$ in terms of the diameter of the output space. This restricts the set of possible $c$-transformed loss function class, thereby allowing us to appeal to standard techniques from empirical process theory to obtain uniform convergence results. Prior work in this area (*e.g.* Lee & Raginsky (2018)) relies on smoothness properties of the loss instead of the geometric properties of the output space, but this precludes non-smooth output metrics.

# B  EXPERIMENTS

All experiments were ran a cluster of CPUS. We do not require a GPU.

## B.1  DATA SETS AND PRE-PROCESSING

**Synthetic**   Synthetic data is generated as described in the main text such that there are 100 queries in the training set and 100 queries in the test set.

**German Credit**   The German Credit data set (Dua & Graff, 2017) consists of 1000 individuals with binary labels indicating if they are credit worthy or not. We use the version of the German Credit data set that Singh & Joachims (2019) used found at `https://www.kaggle.com/uciml/german-credit`. In particular, this version of the Geramn Credit data set only uses the following features: `age` (integer), `sex` (binary, does not include any marital status information unlike the original data set), `job` (categorical), `housing` (categorical), `savings account` (categorical), `checking account` (integer), `credit amount` (integer), `duration` (integer), and `purpose` (categorical). See Dua & Graff (2017) for an explanation of each feature.

Categorical features are the only features with missing data, so we treat missing data as its own category. The following features are standardized by subtracting the mean and dividing by the standard deviation (before this data is turned into LTR data): `age`, `duration`, and `credit amount`. The remaining binary and categorical features are one hot encoded.

We use an 80/20 train/test split of the original 1000 data points, and then sample from the training/testing set with replacement to build the LTR data as discussed in the main text. For our experiments, we use 10 random train/test splits.

**Microsoft Learning to Rank** The Microsoft Learning to Rank data set (Qin & Liu, 2013) consists of query-web page pairs each of which has 136 features and integral relevance scores in $[0, 4]$. We use Fold 1's train/validation/test split. Following Yadav et al. (2019), we use the data in Fold 1 and adopt the given train/validation/test split. The data and feature descriptions can be found at `https://www.microsoft.com/en-us/research/project/mslr/`. We remove the `QualityScore` feature (feature 132) since we use the `QualityScore2` (feature 133) feature to learn the fair metric, and it appears based on the description of these features, they are very similar. We standardize the remaining features (except for the features corresponding to `Boolean model`, i.e. features 96-100, which are binary) by subtracting the mean and dividing by the standard deviation. Following Yadav et al. (2019), we remove any queries with less than 20 web pages. Furthermore, we only consider queries that have at least one web page with a relevance of 4. For each query, we sample 20 web pages without replacement until at least one of the 20 sampled web pages has a relevance of 4. After pre-processing, there are 33,060 train queries, 11,600 validation queries, and 11,200 test queries.

## B.2 COMPARISON METRICS

Let $r$ be a ranking (i.e. permutation) of a set of $n$ items that are enumerated such that $r(i) \in [n]$ is the position of the $i$-th item in the ranking and $r^{-1}(i) \in [n]$ is the item that is ranked $i$-th. Let $\text{rel}_q(i)$ be the relevance of item $i$ given a query $q$.

**Normalized Discounted Cumulative Gain (NDCG)** Let $S_n$ be the set of all rankings on $n$ items. The discounted cumulative gain (DCG) of a ranking $r$ is

$$\text{DCG}(r) = \sum_{i=1}^{n} \frac{2^{\text{rel}_q(r^{-1}(i))} - 1}{\log_2(i+1)}.$$

The NDCG of a ranking $r$ is

$$\frac{\text{DCG}(r)}{\max_{r' \in S_n} \text{DCG}(r')}.$$

Because we learn a distribution over rankings and the number of rankings is too large, we cannot compute the expected value of the NDCG for a given query. Thus, for each query in the test set, we sample $N$ rankings (where $N = 10$ for synthetic data, $N = 25$ for German credit data, and $N = 32$ for Microsoft Learning to Rank data) from the Placket-Luce distribution, compute the NDCG for each of these rankings, and then take an average. We refer to this quantity as the *stochastic NDCG*.

**Kendall's tau correlation** Let $r$ and $r'$ be two rankings on $n$ items. Then

$$\text{KT}(r, r') := \frac{1}{\binom{n}{2}} \sum_{\{i < j : i, j \in [n]\}} \text{sign}(r(i) - r(j)) \text{sign}(r'(i) - r'(j))$$

is the Kendall's tau correlation between two rankings.

**(Disparity of) Group exposure** This definition was first proposed by Singh & Joachims (2019). Assume each item belongs to one of two groups. Let $G_1$ (respectively $G_0$) be the set of items for a query $q$ that belongs to group 1 (respectively group 0). For $i \in \{0, 1\}$, let $M_{G_i} = \frac{1}{|G_i|} \sum_{d \in G_i} \text{rel}_q(d)$, which is referred to as the merit of group $i$ for query $q$. For a ranking $r$ and for $i \in \{0, 1\}$, let $v_r(G_i) = \frac{1}{|G_i|} \sum_{d \in G_i} \frac{1}{\log_2(r(d)+1)}$. Because we learn a distribution over rankings and the number of rankings is too large, we cannot compute the expected value of $v_r(G_i)$ over this distribution. Instead, we sample $N$ rankings (where again $N = 10$ for synthetic data, $N = 25$ for German credit data, and $N = 32$ for Microsoft Learning to Rank data) from the Placket-Luce model. Let $R_q$ be the set of these $N$ sampled rankings for query $q$. Then the stochastic disparity of group exposure for query $q$ is

$$
\begin{cases}
\max \left\{ 0, \dfrac{\frac{1}{N} \sum_{r \in R_q} v_r(G_0)}{M_{G_0}} - \dfrac{\frac{1}{N} \sum_{r \in R_q} v_r(G_1)}{M_{G_1}} \right\} & \text{if } M_{G_0} \geq M_{G_1} > 0 \\
\max \left\{ 0, \dfrac{\frac{1}{N} \sum_{r \in R_q} v_r(G_1)}{M_{G_1}} - \dfrac{\frac{1}{N} \sum_{r \in R_q} v_r(G_0)}{M_{G_0}} \right\} & \text{if } 0 < M_{G_0} < M_{G_1} \\
0 & \text{if } M_{G_0} = 0 \text{ or } M_{G_1} = 0.
\end{cases}
$$

In the language of Singh & Joachims (2019), we use the identity function for merit, and set the position bias at position $j$ to be $\frac{1}{\log_2(1+j)}$ just as they did.

## B.3 SENSTIR IMPLEMENTATION DETAILS

We implement SenSTIR in TensorFlow and use the Python `POT` package to compute the fair distance between queries and to compute Equation (3.4), which requires solving optimal transport problems. Throughout this section, variable names from our code are italicized, and the abbreviation we use to refer to these variables/hyperparameters are followed in parenthesis.

**Fair regularizer optimization**  Recall that in all of the experiments, the fair metric $d_{\mathcal{X}}$ on items is the Euclidean distance of the data projected onto the orthogonal complement of a subspace. In order to optimize for the fair regularizer in Equation (SenSTIR), first we optimize over this subspace, and we refer to this step as the *subspace attack*. Note, the distance between the original queries and the resulting adversarial queries in the subspace is 0. Second, we use the resulting adversarial queries in the subspace as an initialization to the *full attack*, i.e. we find adversarial queries that have a non-zero fair distance to the original queries. We implement both using the Adam optimizer (Kingma & Ba, 2015).

**Learning rates**  As mentioned above, we use the Adam optimizer to optimize the fair regularizer. For the subspace attack, we set the learning rate to *adv_step*($as$) and train for *adv_epoch*($ae$) epochs, and for the full attack, we set the learning rate to *l2_attack*($fs$) and train for *adv_epoch_full*($fe$) epochs. We also use the Adam optimizer with a learning rate of .001 to learn the parameters of the score function $h_\theta$.

**Fair start**  Our code allows training the baseline (i.e. when $\rho = 0$) for a percentage–given by *fair_start*($frs$)–of the total number of epochs before the optimization includes the fair regularizer.

**Using baseline for variance reduction**  Following Singh & Joachims (2019), in the gradient estimate of the empirical version of $\mathbb{E}_{q\sim Q}\big[U(\pi \mid q)\big]$ in Equation (SenSTIR), we subtract off a baseline term $b(q)$ for each query $q$, where $b(q)$ is the average utility $U(\pi \mid q)$ over the Monte Carlo samples for the query $q$. This counteracts the high variance in the gradient estimate (Williams, 1992).

**Other hyperparameters**  In Tables 1 and 2, $E$ stands for the total number of epochs used to update the score function $h_\theta$, $B$ stands for the batch size, $l2$ stands for the $\ell_2$ regularization strength of the weights, and $MC$ stands for the number of Monte Carlo samples used to estimate the gradient of the empirical version of $\mathbb{E}_{q\sim Q}\big[U(\pi \mid q)\big]$ in Equation (SenSTIR) for each query.

## B.4 HYPERPARAMETERS

For the synthetic data, we use one train/test split. For the German experiments, we use 10 random train/test splits all of which use the same hyperparameters. For the Microsoft experiments, we pick hyperparameters on the validation set (where the range of hyperparameters considered are reported below) based on the trade-off of stochastic NDCG and individual (respectively group) fairness for SenSTIR (respectively Fair-PG-Rank), and report the comparison metrics on the test set.

**Fair metric**  For the synthetic data experiments, we use `sklearn`'s logistic regression solver to classify majority and minority individuals with $1/100$ $\ell_2$ regularization strength. For German and Microsoft, we use `sklearn`'s RidgeCV solver with the default hyperparameters to predict age and quality web page score, respectively. For the German experiments, when predicting age, each individual is represented in the training data exactly once, regardless of the number of queries that an individual appears in.

**SenSTIR**  For every experiment, all weights are initialized by picking numbers in $[-.0001, .0001]$ uniformly at random, $\lambda$ in Algorithm 1 is always initialized with 2, and the learning rate for Adam for the score function $h_\theta$ is always .001. For synthetic data, the fair regularization strength $\rho$ varied in $\{.0003, .001\}$. For German, $\rho$ is varied in

$\{.001, .01, 0.02, 0.03, 0.04, 0.05, 0.06, 0.06, 0.07, 0.08, 0.09, .1, 0.11, 0.12, 0.13, 0.14, 0.15, 0.16,$
$0.17, 0.18, 0.19, 0.28, 0.37, 0.46, 0.55, 0.64, 0.73, 0.82, 0.91, 1, 2, 3, 4, 5, 6, 7, 8, 9, 10, 50, 100\}$. For
Microsoft, $\rho$ is varied in $\{.00001, .0001, .001, .01, .04, .07, .1, .33, .66, 1.\}$. We report results for all
choices of $\rho$.

See Table 1 for the remaining values of hyperparameters where the column names have been defined
in the previous section except for $\epsilon$, which refers to $\epsilon$ in the definition of the fair regularizer. For
Microsoft, the best performing hyperparameters on the validation set are reported where the $\ell_2$
regularization parameter for the weights are varied in $\{.001, .0001, 0\}$, $as$ is varied in $\{.01, .001\}$, $ae$
and $fe$ are varied in $\{20, 40\}$, and $\epsilon$ is varied in $\{1, .1, .01\}$.

Table 1: SenSTIR hyperparameter choices

|  | $E$ | $B$ | $as$ | $ae$ | $\epsilon$ | $fs$ | $fe$ | $frs$ | $l2$ | $MC$ |
|---|---|---|---|---|---|---|---|---|---|---|
| Synthetic | 2K | 1 | 0.001 | 20 | 0.001 | 0.001 | 20 | 0 | 0 | 10 |
| German | 20K | 10 | .01 | 20 | 1 | 0.001 | 20 | .1 | 0 | 25 |
| Microsoft | 68K | 10 | .01 | 40 | .01 | 0.001 | 40 | .1 | 0.001 | 32 |

**Baseline and Project**  For the baseline (i.e. $\rho = 0$ with no fair regularization) and project baseline,
we use the same number of epochs, batch sizes, Monte Carlo samples, and $\ell_2$ regularization as in
Table 1 for SenSTIR. Furthermore, we use the same weight initialization and learning rate for Adam
as in the SenSTIR experiments.

**Fair-PG-Rank**  We use the implementation found at https://github.com/ashudeep/
Fair-PGRank for the synthetic and German experiments, whereas we use our own implementa-
tion for the Microsoft experiments because we could not get their code to run on this data. They
use Adam for optimization, and the learning rate is .1 for the synthetic data and .001 for Ger-
man and Microsoft. Let $\lambda$ refer to the Fair-PG-Rank fair regularization strength. For synthetic,
$\lambda = 25$. For German, $\lambda$ is varied in $\{.1, 1, 1.5, 2, 2.5, 3, 3.5, 4\}$. For Microsoft, $\lambda$ is varied in
$\{.001, .01, .1, .5, 1, 2, 3, 4, 5, 6, 7, 8, 9, 10, 50, 100, 500, 150, 200, 250, 300, 350, 400, 450, 500, 550,$
$600, 650, 700, 750, 800, 850, 900, 950, 1000\}$. We report results for all choices of $\lambda$. See Table 2
which summarizes the remaining hyperparameter choices.

Table 2: Fair-PG-Rank hyperparameter choices

|  | $E$ | $B$ | $l2$ | $MC$ |
|---|---|---|---|---|
| Synthetic | 5 | 1 | 0 | 10 |
| German | 100 | 1 | 0 | 25 |
| Microsoft | 68K | 10 | .01 | 32 |

