# OpenReview forum: "Individually Fair Rankings"
_ICLR.cc/2021/Conference — ICLR 2021 Poster_

### Official Review · AnonReviewer1 · 2020-10-27
**Weak accept**

**Rating:** 5
**Confidence:** 3

**Review:**

The paper address the problem of fair ranking. The authors use a notion of individual fairness, meaning that two similar inputs should receive similar outputs. The presented method uses a transport based regularizer to reach fairness.
The authors present a new Algorithm SenSTIR and test it on a synthetic and two real world datasets


The paper is nicely written and readable, however the maths are hard to follow and little intuition is given.
I didn't find an explanation of how the counterfactuals are computed. This is a difficult task that need many assumptions and hypotheses, just "flipping" the gender is hardly enough as gender is correlated to all others attributes.
The legend of figure 3 and 4 are not the same "Kendall's Tau Correlation with Counterfactuals" and "Counterfactual Kendall's Tau Correlation", is it the same metric or something changed ?

In figure 2, Fair-PG-Rank seems to perform very poorly, worse than baseline. As it is the state of the art method, I think some explanation of why it is the case are needed.

Finally most of the paper is based on work done in Yurochkin &Sun 2020 and Yurochkin et al 2020. I think that the authors should explicit the link between the current paper and those two citations. I also think that the main paper should be self content and that Theorem 2.3 of Yurochkin &Sun 2020 should be written down with its hypotheses.

---- After reading answers ------

I'm still not convinced by the computation of counterfactuals, I still disagree with "flipping" the private attribute, because then the naive solution of removing them from the input of the model would appear perfectly fair. This approach seems to forgot about potential proxies, that are not the private attribute, but that a model can learn on and be biases.
Moreover, if I understand correctly, the same computation of counterfactuals is used for the model and the test set. This is unfair as SenSTIR is the only algorithm to use the same kind of counterfactual data than the one used for the evaluation.

I think this is a crucial point, I prefer to lower my rating to 5.

---

> ### Author Response · Authors · 2020-11-17
> **Thank you for your feedback**
>
> Thank you for your feedback. To make our paper better self-contained, we included the precise statement of Theorem 2.3 of Yurochkin & Sun 2020. We address your questions and comments below.
>
> **However the maths are hard to follow and little intuition is given. I didn't find an explanation of how the counterfactuals are computed.**
>
> There are two places where counterfactuals are used: SenSTIR and the evaluation metric for individual fairness in the experiments.
>
> Regarding SenSTIR and counterfactuals, Line 3 in Algorithm 1 is where the counterfactuals are computed in SenSTIR. For each query $q_{t_i}$ in the minibatch, we find its counterfactual $q_{t_i}'$ by maximizing the given optimization problem. This optimization problem is the dual of the fair regularizer in eq. 2.4 (please see a discussion after eq. 2.4 for the intuition behind the regularizer).
>
> To evaluate individual fairness in our experiments, we quantify if the LTR system produces similar ranking distributions for a query and its counterfactual. For the German experiments, for every test query, we create a counterfactual that consists of the same set of people as the test query except each person's gender is flipped. For the Microsoft learning-to-rank experiments, for every test query $q'$, we find the closest query $q$ in the training and testing set such that $q' \neq q$ and where the distance is with respect to the fair metric on queries.
>
> **Just "flipping" the gender is hardly enough as gender is correlated to all others attributes.**
>
> First, this example we use throughout the paper to explain SenSTIR at a high-level is simplistic for the sake of illustration. In general and in our real data experiments, the counterfactual queries that SenSTIR uses in Algorithm 1 can differ from the original query in many ways--not just in one feature--depending on the fair metric. Second, we evaluate the individual fairness properties of the German credit data experiments with *gender* flipped counterfactuals for the sake of intepretability. This evaluation is non-trivial since the fair metric directly deals with *age* information, not gender, so we see how individual fairness can generalize from age to gender. We imagine that ML systems can be unfair to people with respect to features that can be difficult to know before deploying these systems, so although flipping gender is a simplistic choice, it illustrates that SenSTIR can be meaningfully individually fair with respect to these potentially unknown features that were not directly considered when learning the fair metric.
>
> **The legend of figure 3 and 4 are not the same "Kendall's Tau Correlation with Counterfactuals" and "Counterfactual Kendall's Tau Correlation", is it the same metric or something changed ?**
>
> The metric is the same in both cases. Thank you for pointing this out. We have updated the plots.
>
> **In figure 2, Fair-PG-Rank seems to perform very poorly, worse than baseline. As it is the state of the art method, I think some explanation of why it is the case are needed.**
>
> Figure 2 shows that Fair-PG-Rank does not necessarily produce individually fair LTR systems. Fair-PG-Rank was made to achieve group--not individual--fairness in LTR systems, so this finding is not surprising.
>
> **Finally most of the paper is based on work done in Yurochkin & Sun 2020 and Yurochkin et al 2020. I think that the authors should explicit the link between the current paper and those two citations.**
>
> We have already commented on the connection between SenSTIR and Yurochkin & Sun 2020 and Yurochkin et al. 2020 in the main text. See the paragraph after Equation 2.4. Please let us know if there is something more specific you would like us to elaborate on. To summarize, we cannot simply apply these prior works in the LTR setting because of two non-trivial challenges: (i) defining a fair distance on queries, i.e., sets of items, and (ii) ensuring the resulting optimization problem is differentiable. Regarding the first challenge, recall that individual fairness requires that similar inputs be treated similarly. Similarity of inputs is quantified by a fair metric. In the LTR setting, the inputs are queries, i.e. sets of items, so we must define a meaningful distance on sets. This challenge is resolved at the end of Section 2.1. Regarding the second challenge, we need to ensure the resulting optimization problem, which depends on the fair metric on queries, is differentiable to make learning possible. This challenge is resolved in Section 3.

---

> ### Author Response · Authors · 2021-03-19
> **Clarification After Reviewer Update: We do not use same counterfactuals for train and test. Train uses only age and test uses only gender.**
>
> In the German credit experiments, we do *not* use the same counterfactuals for training and testing in order to more fairly evaluate SenSTIR. During training, we only use *age* counterfactuals, whereas during testing, we only use *gender* counterfactuals. This is a way of assessing *exactly* what you were suggesting: whether the resulting ranking policy is robust to changes in proxy variables. Gender is not treated as a private (i.e. protected) attribute during training. Please see our original response for more details.

---

### Official Review · AnonReviewer4 · 2020-10-28
**Individually Fair Rankings.  Interesting paper that extends the individual fairness approach to learning to rank domain.**

**Rating:** 7
**Confidence:** 3

**Review:**

The authors extended individual fairness approach to the domain of learning to rank domain. This paper proposes a method for training individually fair learning to rank models by making use of optimal transport-based regularizer.

This work appears to be an extension to Yurochkin et. al. 2020 ICLR SenSR paper in which fair metric was learned from data. While that papers focused on training individually fair ML models by casting the problem as training supervised ML models that are robust to sensitive perturbations, this paper extended the idea to individual fair ranking that are invariant to sensitive perturbations of features of ranked items.

###############
This paper is well written. The code and the datasets used for the experimentation have been provided.
I vote to accept this paper. I like the approach presented in this paper for solving this very important issue.

###############
The importance of the problem of dealing with bias in search results cannot be understated. Example of the issue of fairness in search ranking: how to ensure that minority candidates (or candidates of opposite gender) would be fairly ranked together with other job candidates when having similar qualifications. This paper attempts to find a solution to this and similar type-problems.


###############
In this paper both theoretical and empirical results were presented. The authors showed that using their optimal transport-based regularizer leads to certified individual fair LTR models.
The results were demonstrated on several datasets: synthetic and two publicly available datasets. Results showed that authors’ method exhibited ranking stability with respect to sensitive perturbations when compared with a state-of-the-art fair LTR model.
Importantly, the authors showed empirically that individual fairness implied group fairness by not vice versa.

###############
Recommendations:
On page 4 section 3 authors reference Theorem 2.3 but in the citation there is no Theorem 2.3.  In the citation there is a relevant equation 2.3 and a note that states this equation comes from another paper. So I believe that this should probably needs to be cited differently (by citing Blanchet & Murthy original paper - assuming they were the first to prove this theorem)
Additionally, authors keep citing Yurochkin & Sun ArXiv pre-paper even through it has already appeared at ICLR 2020.

Question:
My only question for the authors is how easy would it be to learn fair metric in a typical application for LTR models?

---

> ### Author Response · Authors · 2020-11-17
> **Thank you for your feedback**
>
> Thank you for your feedback to improve our paper, and we fixed the bibliography as you pointed out. We address your question and comment below.
>
> **My only question for the authors is how easy would it be to learn fair metric in a typical application for LTR models?**
>
> One major contribution of our work is showing how to obtain a fair metric on queries--needed in order to use SenSTIR--from a fair metric on item representations via optimal transport tools. See the end of Section 2.1. In our real data experiments, we showed that it is possible to learn effective fair metrics in the LTR setting based off the prior techniques for fair metric learning designed for regression/classification (Yurochkin et al. 2020, Mukherjee et al. 2020). Hence, in the future, SenSTIR can utilize new methods that are not necessarily intended for the LTR setting for learning fair metrics.
>
> **On page 4 section 3 authors reference Theorem 2.3 but in the citation there is no Theorem 2.3.**
>
> Theorem 2.3 of "SenSeI: Sensitive Set Invariance for Enforcing Individual Fairness" by Yurochkin & Sun 2020 (https://arxiv.org/pdf/2006.14168.pdf) can be found at the top of page 4 in their paper. We have also added the statement of this theorem to section 3 of our paper for completeness.

---

### Official Review · AnonReviewer3 · 2020-10-28
**Interesting paper with some major flaws**

**Rating:** 4
**Confidence:** 4

**Review:**

The paper tackles the problem of providing rankings that are "fair" to individual items being ranked as opposed to groups of items as in previous work on fair ranking.

I have three major issues with the paper:

1- A practical objection to the approach is the use of randomized rankers to get around the important issue that ranking is by its very nature unfair to individuals. To take the example of job applications used in the paper, if 100 people are applying to one job and a ranked list of the candidates is presented to the person making the hiring decision, then well one person will be at the top and one person at the bottom. Now, if there are 100 job openings and the same 100 people apply to all of them, then yes randomizing the results can deal with this issue, but then fairness is not even an issue because everyone gets a job and there won't be any complaints. In other words, fairness is only an issue is there is a lack of resources, which means that randomization doesn't really solve the issue, but just gives one the illusion that it is resolved. I feel like the authors haven't spent too much time thinking about these issues and that they're mostly thinking about the unbounded query scenario. But, even in that setting, there are very few products where the people in charge would be willing to randomize the ranked items because that would result in degradation in user experience (i.e. in this example, the person doing the hiring). There have been some recent attempts to address the planning issue that comes up when dealing with this, so I'd suggest for the authors to have a look at papers like this one: https://arxiv.org/abs/2008.00104

2- A more fundamental issue is that I just can't find a proper metric for individual fairness in the paper. The only proposal is a comparison between the outputs of the ranker before and after altering some features fed into it. This way of evaluating rankers makes no sense to me at all because a really easy way for a ranker to appear fair is by not using the feature being modified (e.g. age or gender) directly, but inferring it from the remaining features. I think the issue is that the authors are confused by the subtle but crucially important distinction between losses and metrics. You're allowed to use whatever information you like when you're designing your loss, but the final evaluation should not rely on the model, but just its output.

3- Finally, on the MSLR dataset, it's important to include LambdaMART as a baseline to report the degradation in NDCG incurred as a result of the fairness constraints. I should point out that there are multiple implementations of LambdaMART out there and it's important to used the best one, which is the one included in the LightGBM package. You can find some results for the MSLR dataset in this paper among others: https://dl.acm.org/doi/pdf/10.1145/3336191.3371844

Given these issues, I don't think the paper is ready to be published and I encourage the authors to think carefully about their problem definition and experimental setup before resubmitting the paper.

---

> ### Author Response · Authors · 2020-11-17
> **Thank you for your feedback. However, some of the criticism is due to misunderstandings. Part 1.**
>
> We thank you for your feedback, and address your comments below.
>
> **A practical objection to the approach is the use of randomized rankers to get around the important issue that ranking is by its very nature unfair to individuals.**
>
> First, stochastic ranking policies are a way to make the LTR problem feasible. Otherwise, searching for utility-maximizing rankings would take time that is exponential in the query size. This is also something that is standard in the literature on fair ranking (see Biega et al 2018, Singh & Joachims 2018, 2019, and Zehlike & Castillo 2020).
>
> Second, we wish to point out that when we evaluate the individual fairness of our ranking policies, we are not averaging over many draws from the ranking policy. In particular, when we evaluate individual fairness by comparing the distance between the ranking of a query and the ranking of its counterfactual query, we use the deterministic rankings given by sorting the items by their predicted scores, instead of drawing rankings from the learned distributions.
>
> Third, while the long-term effects of ranking policies are interesting, they are difficult to study because the results are very sensitive to the exact model for how the ranking policy affects the state of the system. This is also a very different problem than the problem that we address in this paper. We now mention the paper you referenced in the conclusion.
>
> **I just can't find a proper metric for individual fairness in the paper. The only proposal is a comparison between the outputs of the ranker before and after altering some features fed into it.**
>
> This statement is not true. We only use this evaluation for the German credit experiments. We evaluate individual fairness in the synthetic and the Microsoft LTR experiments quite differently as stated in the second paragraph of the "Results" paragraph of Section 5.1 and in the "Comparison metrics" paragraph in Section 5.3. In particular, for the Microsoft experiments, for every query q in the test set, we find the closest query $q'$ in the train and test set such that $q' \neq q$ where the distance is with respect to the fair metric. Then we compare the outputs of the ranker on $q$ and $q'$. We can think of $q'$ as a counterfactual that exists in the data.
>
> **This way of evaluating rankers makes no sense to me at all because a really easy way for a ranker to appear fair is by not using the feature being modified (e.g. age or gender) directly, but inferring it from the remaining features. I think the issue is that the authors are confused by the subtle but crucially important distinction between losses and metrics. You're allowed to use whatever information you like when you're designing your loss, but the final evaluation should not rely on the model, but just its output.**
>
> We think that the reviewer misunderstood the evaluation metrics. We agree that the final evaluation should not rely on the model - and that is what we did in our evaluation when flipping the gender feature in the German experiments, a feature not given special consideration when training SenSTIR or learning a fair metric. We provide a more detailed justification of this choice below.
>
> We use this evaluation metric on the German credit experiments since it is (1) easily interpretable, (2) non-trivial, and (3) is standard in the literature. First, individual fairness means that pairs of similar inputs, i.e. queries, should be treated similarly. One interpretable way to check this notion of fairness is to see if two queries of people that differ only with respect to gender are ranked similarly.
>
> Second, this evaluation metric is non-trivial for the German credit experiments since the fair metric is trained to ignore variation in the data due to *age*, but we evaluate individual fairness with respect to *gender*. We imagine that ML systems can be unfair to people with respect to features that can be difficult to know before deploying these systems, so although flipping gender is a simplistic choice, it illustrates that SenSTIR can be meaningfully individually fair with respect to these potentially unknown features that were not given special consideration when choosing the fair metric or in training with SenSTIR. In addition, the "Project" baseline (see the "Baselines" paragraph in the beginning of Section 5) does not use age as a feature yet it is significantly not individually fair with respect to counterfactual queries obtained by flipping the gender of each person.
>
> Third, our evaluation metric for individual fairness in the German credit experiments is the natural analogue of what is standard in the individual fairness literature for classification (Yurochkin et al. 2020, Yurochkin & Sun 2020, Garg et al. 2018). We are open to suggestions of other evaluation metrics.

---

> > ### Author Response · Authors · 2020-11-17
> > **Thank you for your feedback. However, some of the criticism is due to misunderstandings. Part 2.**
> >
> > **Comparison to LambdaMART**
> >
> > First, the point of our paper is not to beat the state of the art LTR algorithm, but to demonstrate how to achieve individual fairness in the LTR setting since prior work has mostly focused on group fair LTR systems. Nonetheless, Singh & Joachims 2019 have experiments demonstrating that stochastic policy rankers (i.e. SenSTIR when the fair regularization $\rho = 0$) produces rankers that are competitive with other LTR baselines.
> >
> > Second, LambdaMART is a boosted tree-based method, so neither our method nor prior work like Singh & Joachims 2019 is applicable since these methods require a differentiable objective. However, we think that coming up with an individually fair version of LambdaMART is an interesting line of future work.

---

### Official Review · AnonReviewer2 · 2020-10-31
**solid paper**

**Rating:** 7
**Confidence:** 2

**Review:**

The paper uses an optimal-transport approach to ensure that two similar items that differ only on a sensitive feature (e.g., gender) are ranked similarly. In this way, individual items are treated fairly. For example, if two job candidates differ in gender but are otherwise similar, the job search engine should rank them similarly. The authors employ an optimization problem they call SenSTIR where the objective is maximizing expected utility minus a regularization term, and where the fairness properties are encoded in the regularization term. The regularization term is weighted by rho, a parameter than increases fairness as it increases. The regularization term itself encourages similar queries to lead to similar rankings. The authors appeal to duality to derive a way to run stochastic optimization on their problem. They prove one theorem that I don't fully understand. Then they test their method on three data sets: simulated data used in related work, German credit data used in prior work, and Microsoft learning to rank data. They compare to training without fairness, a projection method, Fair-PG-Rank, and random. They try different values of rho, showing the larger rho indeed leads sensibly to more individually fair rankings. They use the NDCG metric for accuracy and Kendall's tau correlation for fairness. They show that they get individual fairness by design and group fairness "for free". In contrast, Fair-PG-Rank design for group fairness does poorly on individual fairness.

The paper is well written and includes an excellent and extensive survey of related work. The authors approach seems reasonable and their analysis and experiments are convincing. I especially like Figure 2 -- this is a nice visualization of individual fairness and was very helpful.

One quibble I had is that the authors said that individual fairness is sufficient for group fairness, or at least "can be sufficient". Usually "sufficient" is a theoretical statement that always holds. Later, it's clear that this is an empirical finding on the particular data sets tried. I'd like the authors to be more careful about using the word "sufficient".

One question I had is how this would generalize to multiple sensitive features? To be practical, that generalization will be valuable.

Can the authors say something about computational complexity? Is the problem 2.4 NP-hard? Is Algorithm 1 an approximation? What is the running time of Algorithm 1 both in theory and in the experiments (seconds). How does the running time of SenSTIR compare with Fair-PG-Rank?

A higher level question is why the authors did not test fair supervised learning algorithms. Yes, as the authors point out, supervised learning is not meant for LTR (and SL doesn't define a fair distance on sets of items and uses different metrics, for example, L2 loss versus NDCG), but a supervised learning method can produce a relevance score. So the experiments could try one of these approaches, and probably should. (I didn't quite understand the point about not defining a distance on sets.)

Minor note:

References: Some of the references should be cleaned up and made more uniform. For example, the reference "Ke Yang, Vasilis Gkatzelis, and Julia Stoyanovich" repeats the conference title four times.

---

> ### Author Response · Authors · 2020-11-17
> **Thank you for your feedback**
>
> Thank you for your feedback. We replaced the word “sufficient” with "adequate" and cleaned-up the references. We address your comments and questions below.
>
> **One question I had is how this would generalize to multiple sensitive features?**
>
> SenSTIR enforces individual fairness, which depends on a fair metric instead of protected groups (indicated by one or many sensitive features). It is certainly possible to define a fair metric in a way that encodes the intersectional invariances from multiple protected groups/sensitive features (e.g. Yurochkin et al. 2020, Mukherjee et al. 2020), and SenSTIR will enforce these invariances.
>
> **Can the authors say something about computational complexity? Is the problem 2.4 NP-hard? Is Algorithm 1 an approximation? What is the running time of Algorithm 1 both in theory and in the experiments (seconds). How does the running time of SenSTIR compare with Fair-PG-Rank?**
>
> Equation 2.4 as written is hard to solve, but the dual is solvable for convex losses. However, if the ranking policies $\pi$ are parameterized by a neural net, then in general problem 2.4 is non-convex. Finding the optimum of non-convex problems is typically NP-hard, yet solving non-convex problems like those in deep learning is still tremendously valuable and practically feasible.
>
> For the German experiments, SenSTIR takes 19.2 minutes, and the average time for each epoch of Algorithm 1 is 0.06 seconds with a standard deviation of 0.016 seconds. Using the implementation of Fair-PG-Rank provided by the authors (found at https://github.com/ashudeep/Fair-PGRank), Fair-PG-Rank takes 73.31 minutes, and the average time for each epoch is 43.98 seconds with a standard deviation of 5.61 seconds. We believe that the run times of both of these algorithms can be significantly improved with optimizations to the code.
>
> **A higher level question is why the authors did not test fair supervised learning algorithms. (I didn't quite understand the point about not defining a distance on sets.)**
>
> At a high level, individual fairness says that an ML system should produce similar outputs for similar inputs (where similarity is measured by the fair metric). As you noted, measuring the similarity of outputs in SL can be generalized to LTR (i.e. measuring the similarity of relevance scores). However, measuring the similarity of inputs in LTR is different. The inputs to the LTR problem are sets of items (i.e. unordered collections of feature vectors), while in SL the inputs are singleton feature vectors. In other words, individual fairness for LTR should intuitively imply similar ranking outputs on similar sets of items, therefore requiring a fair metric on sets. In other words, it is the set, not individual items that matter here. This difference prevents us from being able to apply fair supervised learning methods directly. Another important difference between LTR and SL is the loss function, as you noted.

---

### Author Response · Authors · 2020-11-17
**General Response**

We thank all the reviewers for their thoughtful comments. We answer each reviewer’s questions individually, and we have updated the draft according to the feedback and suggestions.

---

### Decision · Program_Chairs · 2021-01-07
**Final Decision**

**Decision:**

Accept (Poster)

**Comment:**

The paper focuses on individual fair ranking and proposes an approach for that based on optimal transport. The reviewers are in general positive about the paper, however, there are a a couple of concerns that I believe should be addressed before publication.

First, I find the treatment of the term "counterfactual" misleading in the paper.  Counterfactual fairness has been proposed in the literature as a causal notion of individual fairness. However, as far as I can see in the paper, there is not such a causal treatment of counterfactuals in the paper. Thus, I suggest the authors to reconsider their treatment of counterfactuals in the paper, as it may trigger confusion. Second, I also agree with R1 that  it is unfair as SenSTIR is the only algorithm to use the same kind of "counterfactual" data than the one used for the evaluation.

---

> ### Author Response · Authors · 2021-03-19
> **Thank you for your time reviewing our work and for your suggestions.**
>
> Thank you for your time reviewing our work and for your suggestions. In the camera-ready version of our paper, we replace all instances of “counterfactual” with “hypothetical.”
>
> For the future readers of this thread we wish to clarify the evaluation of individual fairness in the German Credit experiment. In this experiment, we do *not* use the same “hypotheticals” (counterfactuals) for training and testing in order to more fairly evaluate SenSTIR. During training, we only use *age* hypotheticals, whereas during testing, we only use *gender* hypotheticals. This procedure is verifying that the resulting ranking policy is robust to changes in proxy variables as R1 was suggesting. We have improved the exposition in our paper to emphasize this fact. Please see our original response to R1 for more details.